# Biobased Carbon Dots: From Fish Scales to Photocatalysis

**DOI:** 10.3390/nano11020524

**Published:** 2021-02-18

**Authors:** Carlotta Campalani, Elti Cattaruzza, Sandro Zorzi, Alberto Vomiero, Shujie You, Lauren Matthews, Marie Capron, Claudia Mondelli, Maurizio Selva, Alvise Perosa

**Affiliations:** 1Department of Molecular Sciences and Nanosystems, Università Ca’ Foscari di Venezia, Via Torino 155, 30172 Venezia Mestre, Italy; carlotta.campalani@unive.it (C.C.); cattaruz@unive.it (E.C.); sandro.zorzi@unive.it (S.Z.); alberto.vomiero@unive.it (A.V.); selva@unive.it (M.S.); 2Division of Material Science, Department of Engineering Sciences and Mathematichs, Luleå University of Technology, 971 87 Luleå, Sweden; shujie.you@ltu.se; 3The European Synchrotron Radiation Facility, 38043 Grenoble CEDEX 9, France; lauren.matthews@esrf.fr (L.M.); marie.capron@esrf.fr (M.C.); 4Partnership for Soft Condensed Matter PSCM, ESRF The European Synchrotron Radiation Facility, 71 Avenue des Martyrs, 38043 Grenoble CEDEX 9, France; 5CNR-IOM, Institut Laue Langevin, 71, Avenue des Martyrs, 38042 Grenoble CEDEX 9, France; mondelli@ill.fr

**Keywords:** carbon dots, sustainability, bio-sourced functional materials, metal-free photoredox catalysis, circular economy, circular chemistry

## Abstract

The synthesis, characterization and photoreduction ability of a new class of carbon dots made from fish scales is here described. Fish scales are a waste material that contains mainly chitin, one of the most abundant natural biopolymers, and collagen. These components make the scales rich, not only in carbon, hydrogen and oxygen, but also in nitrogen. These self-nitrogen-doped carbonaceous nanostructured photocatalyst were synthesized from fish scales by a hydrothermal method in the absence of any other reagents. The morphology, structure and optical properties of these materials were investigated. Their photocatalytic activity was compared with the one of conventional nitrogen-doped carbon dots made from citric acid and diethylenetriamine in the photoreduction reaction of methyl viologen.

## 1. Introduction

The term “circular economy” is currently recognized as one of the drivers for research in chemistry, as it is the science that holds the key for the recovery of waste and its conversion into useable materials, chemicals and products.

A yet untapped source of biowaste derives from the fish (catch and farming) industries. In fact, in 2016 the worldwide fish production amounted to 171 million tons, where China was the major productor [1]. Furthermore, the average consumption of fish has increased in the past forty years from 12.6 to 19.8 kg per person [2].

The estimated waste from fisheries is still a matter of debate, with yearly quantities varying in a wide range. However, though this scenario is not completely clear, the resulting amount of fish biowaste is in the order of dozens of million tons per year [3]; this could represent a resource of sustainable chemical richness [4]. Fish residues are comprised of whole waste fish, head, viscera, skin, bones, etc., and their composition is variable according to the specie, sex, age, time of the year and geographic area. However, all fish biowaste contains several valuable molecules such as oils, collagen, chitin, gelatin and pigments [5]. Nowadays, the most common option for the treatment of such waste is a low-tech solution for the obtainment of fishmeal: A product used in the animal feed and fertilizer sectors [6,7]. Despite this, in recent years, various processes and technologies have been reported for the upgrade to high added-value products from fishery waste [8,9].

In this work we studied the possibility of valorization of bass (*Dicentrarchus labrax*) scales, using them as the carbon source for the synthesis of Carbon Dots (CDs): Luminescent carbon nanomaterials. Bass is a common fish in the Northern Adriatic Sea and its farming and consumption are well spread in Northern Italy. The choice of using scales as the carbon source was made based on their composition. In fact, they represent an inexpensive, accessible source of nitrogen-containing biomass that does not directly impact the food chain. The principal components of fish scales are collagen, chitin, minerals, lipids and pigments. Chitin is the main organic component and is a natural nitrogen containing polysaccharide (poly-*β*(1/4)-N-acetyl-D-glucosamine) that provides an efficient precursor for the N-doped CDs [10].

Thanks to their superiority in water solubility, chemical inertness, low toxicity, ease of functionalization and resistance to photobleaching, CDs have inspired intensive research efforts in recent years [11,12].

These nanoparticles have drawn considerable attention in a wide range of applications, ranging from biomedical [13,14,15] to energy-related fields [16,17]. Among them, CDs appeared to be a promising luminescent bio-based nanomaterial for photocatalytic applications; however, to date, the photocatalytic activity of CDs has been under-explored despite their remarkable light harvesting properties and excellent electron donor/acceptor capabilities. Moreover, most reported photosystems of CDs are limited by the co-presence of precious metal complexes or enzymes as redox mediators [18,19,20].

The accurate design of CDs with tunable photoelectric properties is one of the major issues faced to achieve a breakthrough in this field. In particular, many efforts have been focused on the doping of CDs with nitrogen and it has been demonstrated that nitrogen doping can significantly enhance their properties and thus expand their applications [21,22].

In one of our previous works, we demonstrated that the carbon source and synthetic procedures diversely affect the structural and optical properties of the CDs, which in turn influence their photoelectron transfer ability [23]. Six different types of CDs, made from citric acid, glucose or fructose and using two different synthetic pathways (hydrothermal or pyrolytic), were tested for the photoreduction of methyl viologen. Overall, citric acid-CDs were found to provide the best photocatalytic performance. This led us to further investigate the citric acid CDs and their photoreduction ability. With this purpose four different CDs were synthesized by using two different methods (hydrothermal and pyrolytic) and two sets of reagents (neat citric acid and a mixture of citric acid and diethylenetriamine). The photocatalytic activity of the nanoparticles was investigated on the model photoreduction of methyl viologen. The most active CDs in the photoreduction was shown to be the amorphous N-doped CDs (a-N-CDs, obtained with hydrothermal treatment using citric acid and diethylenetriamine as reagents) [24]. For this reason, in this work, we decided to compare the new class of carbon nanomaterials from fish scales (bass-CDs) with a-N-CDs for the photoreduction of methyl viologen.

In recent years, the preparation of carbon nanoparticles from biomass is starting to be investigated because biomass carbon sources are widely available, inexpensive, green and safe starting materials. Compared to molecular carbon sources, biomass, and in particular waste biomass, has therefore many advantages [25]. Moreover, biomass naturally contains heteroatoms in contrast with man-made carbon sources that require external addition [26]. In recent literature, various types of biomasses have been used as carbon sources to synthesize CDs, such as walnut peel [27], papaya juice [28], goose feather [29] and bee pollen [30], and they have been applied to various fields: Bio-imaging, sensors, drug delivery and others. Fish scales have also been recently described for the preparation of fluorescent nanomaterials with applications in different fields. Zhang et al. [10], for example, reported a facile synthesis route for the preparation of CDs from grass carp scales for the detection of hypochlorite. Another work shows how fish scale-derived CDs could be used also for the determination of the presence of ferric ions in real water samples and human serum [31]. Carbon dots produced from fishery waste were synthesized also from Kai et al. [32] and used for the detection of pharmaceuticals molecules as lidocaine hydrochloride. Another application of fish scale-derived CDs is in the biomedical field: Alshatwi et al. [33], for example, produce simultaneously hydroxyapatite and carbon nanoparticles from *Lethrinus lentjan* scales applicable for bio-imaging and bone-tissue engineering applications. In this context, however, the photocatalytic activity of this type of CDs is still underexplored.

## 2. Materials and Methods

### 2.1. Materials

All the reagents were purchased from Merck Life Science S.r.l. (Milano, Italy), were of analytical grade and used without further purification. MilliQ water was used as a solvent throughout the experiment and it was obtained with a Merck Millipore C79625 system.

The scales were of sea bass (*Dicentrarchus labrax*) and were purchased from a local market. Before use, the scales were washed thoroughly with water and dried in a vacuum oven at 70 °C overnight. Figure 1 summarizes the unit operations from scales to CDs.

### 2.2. Synthesis of CDs from Fish Scales

In a typical experiment 2 g of dried and grounded fish scales were put in a Teflon-lined autoclave with 20 mL of MilliQ water. The system was heated at 200 °C for 24 h. The obtained brownish suspension was filtered, and the water was removed by rotary evaporation to obtain a brown solid with 30–50% yield.

For some studies the CDs were further purified via dialysis: A total of 200 mg was dispersed in 3 mL of milliQ water and dialyzed for 24 h using a 1KDa dialysis membrane in 2 L milliQ water. The resulted purified CDs were obtained with 2–6% yield.

### 2.3. Characterization of CDs

The physical characterization of the nanoparticles was determined by CHNS elemental analysis, Fourier-Transform Infrared (FT-IR) spectroscopy, X-Ray Photoelectron Spectroscopy (XPS), Small Angle X-Ray Scattering (SAXS) and Static Light Scattering (SLS).

CHNS analysis was performed on an Elemental Unicube (Elementar Italia Srl, Lomazzo, Italy) and FT-IR spectra have been recorded on a Perkin-Elmer Spectrum One FT-IR spectrometer (PerkinElmer Italia, Milano, Italy) at wavenumbers ranging from 400 to 4000 cm^−1^. XPS was performed with a Perkin Elmer Φ 5600ci spectrometer (PerkinElmer Italia, Milano, Italy) in the 10^−7^ Pa pressure range, by using nonmonochromatic Al Kα radiation (1486.6 eV). The binding energy (BE) values are referred to the Fermi level. BE scale calibration was confirmed by the position of both Au4f_7/2_ and Cu2p_3/2_ bands in pure metal samples, falling, respectively, at 84.0 and 932.6 eV. Wide range survey spectra were recorded for all the sample. Single spectra were recorded for C1s, O1s and N1s regions. The recorded XPS bands were fitted using a non-linear least-square fitting procedure adopting a Shirley-type background and Gaussian–Lorentzian peak shapes for all the peaks (XPSPEAK41 free software, 4.1, Raymund W.M. Kwok, Hong Kong, China). The BE correction from the surface charging evidenced during analysis (around 2–3 eV) was done by using as internal reference, with the position at 284.6 eV of the C1s band related to C=C bonds [24], checking, in addition, the consistency of the BE positions of all the other bands evidenced in the different XPS peaks. The BE values final uncertainty was not larger than 0.2 eV. The atomic composition of the analyzed region (about 5–10 nm of thickness from the surface) was estimated by using sensitivity factors provided by Φ V5.4A software (Perkin Elmer, Physical Electronics, Eden Prairie, MN, USA): The relative uncertainty of the atomic fraction of the different elements was lower than 0.1.

SAXS measurements were performed at the European Synchrotron Radiation Facility (ESRF; Grenoble, France) on the ID02 beamline [34], using an X-ray wavelength of ~0.1 nm. Two sample-to-detector distances (SDD) were used (1 and 8 m) to obtain a Q-range of 0.0008–0.74 Å^−1^ (Q = scattering angle). Scattering patterns were collected by taking 10 successive exposures of 0.05 s to reduce radiation damage. The measured two-dimensional scattering patterns were normalized to an absolute intensity scale and azimuthally averaged to obtain the one-dimensional scattering profiles. The corresponding background was then subtracted before the profiles, from different SDDs, were merged. The one-dimensional SAXS profiles were analyzed using the SasView software [35].

Static Light Scattering (SLS) measurements were performed using a Malvern Zetasizer (Malvern Products, Palaiseau, France) with a He-Ne laser light source (633 nm). The measurements were carried out at 25.0 °C for a duration of 30 s for each one, and the samples were dispersed in water, filtered at 0.22 mm and placed in 1 mL cuvettes.

The high-resolution Atomic Force Microscopy (AFM, Oxford Instruments Cypher Asylum Research, available at the AFM platform of the PSCM at Grenoble) was used in tapping mode to investigate dimensions and morphology of the obtained nanoparticles or of their aggregates. The samples were prepared in accordance with the standard procedures using a spin coater.

For the optical characterization of CDs, an UV–Vis spectrophotometer Agilent 8456 (Agilent Technologies Italia, Milano, Italy) and photoluminescence excitation (PLE) were used and emission (PL) spectra were recorded by a Perkin Elmer LS 55 fluorescence spectrophotometer (PerkinElmer Italia, Milano, Italy).

^1^H, ^13^C{^1^H}, DOSY NMR spectra were recorded on a Bruker AV 300 (^1^H: 300 MHz; ^13^C: 75.5 MHz; 51V: 78.28 MHz) spectrometer (Bruker GmbH, Mannheim, Germany). For ^1^H and ^13^C{^1^H} NMR, the chemical shifts (δ) were reported in parts per million (ppm) relative to the residual undeuterated solvent as an internal reference.

### 2.4. Photoreduction of Methyl Viologen (MV)

A solution composed by 0.1 M of EDTA, 60 µM of MV with an absorbance normalized amount of CDs (a.b.s. = 0.13 a.u.; 0.005 mg/mL a-N-CDs, 0.2 mg/mL bass-CDs) were placed under inert atmosphere in a quartz cuvette. The solutions were then irradiated at 365 nm fixed wavelength emission (Hangar s.r.l.; ATON LED-UV 365; 80 W/m^2^ of irradiance in the UVA spectral range 315–400 cm^−1^). The progress of the reactions was monitored using an UV spectrophotometer following the formation of the typical absorption band of the reduced MV^+.^ radical cation form centered at 605 nm, and its concentration was estimated using ε = 13,700 M^−1^ cm^−1^. The same experiment was conducted also using the same concentration of bass-CDs and a-N-CDs (0.2 mg/mL).

A series of blank reaction were also performed to assess the effectiveness of the CDs in catalyzing the reduction of MV (E = −0.45 V vs NHE). All the blank tests performed reveal that CDs is essential to guarantee the effectiveness of the photochemical reduction of the MV.

## 3. Results and Discussion

### 3.1. Synthesis and Characterization of Bass-CDs

All the carbonaceous nanoparticles were synthesized by hydrothermal treatment as previously reported [23,24].

The nanoparticles made from the waste bass scales were obtained in 30–50%wt yields. This variability was probably due to the different composition of the scales depending on season, fish age, sex, habitat, etc. The lower yields, compared with the 72% yield of a-N-CDs, clearly indicate that a high fraction of the biomass is not convertible into CDs. In fact, fish scales also contain a large fraction of minerals such as CaCO_3_ (27–47% [36,37,38]). An average composition of the scales was obtained by elemental analysis as reported in Table 1.

An exploratory test for the removal of minerals from scales was also performed. Bass scales were pre-treated by stirring in 0.25 M HCl for 30 min at room temperature followed by rinsing to neutral pH and drying. The demineralized scales were then employed to synthesize CDs by hydrothermal treatment (200 °C, 24 h). The CDs made from demineralized scales were CHNS analyzed showing a C:N ratio of 3.2; the same as for the bass-CDs obtained from non-demineralized scales. This implies that the minerals present in the fish scales are not involved in the formation of the nanoparticles. Based on this evidence, all the following CDs syntheses were performed without demineralizing the scales. The photocatalytic ability of the CDs obtained after dialysis and of the demineralized ones were compared to that of the CDs made by simple treatment of the scales and showed no significant difference.

When the bass-CDs were dialyzed, the obtained purified yields were 2–6%. The resulting materials were predominantly carbonaceous solids as confirmed by the silent ^1^H, ^13^C {^1^H} and two-dimensional (2D) DOSY NMR spectra (Appendix A). The NMR experiments excluded the presence of soluble molecular and/or oligomeric species. On the other hand, the a-N-CDs were sufficiently small to completely permeate the membrane, probably indicative of a significant formation of molecular-like fluorophores.

In our previous work [23], we showed that the synthetic procedure (hydrothermal vs pyrolytic) determines the morphology of the CDs. Hydrothermal treatment—used in the present case—yields predominantly amorphous CDs lacking a graphitic core.

The morphology, composition and surface properties of the nanoparticles were studied by Small Angle X-Ray Scattering (SAXS), Static Light Scattering (SLS), Fourier Transform InfraRed (FT-IR) spectroscopy, X-ray Photoelectron Spectroscopy (XPS), Atomic Force Microscopy (AFM) and elemental analysis.

The elemental composition of the nanoparticles indicated a C:N ratio of 3.2 for both the bass-CDs and the a-N-CDs used for comparison (Table 1). This result allowed to conclude that the CDs synthesized from bass scales have the same amount of nitrogen as the a-N-CDs, without the need of any doping agent.

The surface morphology of the nanoparticles was evaluated by FT-IR and XPS and the results are summarized in Table 2.

FT-IR analysis (spectrum in Appendix A) showed a broad absorption band around 3400 cm^−1^ corresponding to the O−H vibrations and a peak around 3300 cm^−1^ due to the N–H stretching. The strong absorption bands around 1700−1600 cm^−1^ reflected the presence of carboxylic acid, ketone and amide groups. Multiple signals in the region between 1600–1400 cm^−1^ were assigned to both C–C stretching of aromatic or conjugated double bonds and to the N–H bending of the N-containing groups.

The XPS wide range spectrum of bass-CDs evidenced bands related to the presence of carbon, oxygen, and nitrogen. The quantitative analysis obtained by the intensity of C1s, O1s and N1s single spectra gave a relative composition of 66% carbon, 26% oxygen and 8% nitrogen (for a-N-CDs, the ratio was 70:20:10 of C:O:N). XPS analysis gave a different C/N ratio in respect to CHNS, because this technique refers only to the surface of the nanoparticles, whereas elemental analysis is related to the whole composition of the material. With this in mind, it can be highlighted that for both bass-CDs and a-N-CDs the nitrogen concentration is lower at the surface than in the bulk. In Appendix A, the C1s band, recorded in high-resolution mode, evidences three different contributions centered at 284.6 eV (related to C=C bonds, 60% of total area), at 285.8 eV (related to C=O and/or C=N bonds, 25% of total area) and at 288.0 eV (related to C–O and/or C–N bonds, 15% of total area) [24]. The O1s band is centered around 531.6 eV, in agreement with the presence of carbon-oxygen bonds. As far as nitrogen is concerned, the N1s band (Appendix A) can be well-fitted only by assuming the overlap of two different components, having approximately the same intensity, centered at 399.8 and at 401.1 eV of binding energy. The first component is in agreement with the presence of nitrogen involved in amine functional groups, the second one in amide compounds [39].

SAXS measurements showed the presence of clusters with dimensions in the order of hundreds of nanometers, from the upturn in intensity in the low Q region of the scattering curves of both samples (Appendix A). The surface fractal dimensions (D_s_), extracted from the fitting procedures (described in SI), informs about the roughness of the clusters surface. The D_s_ values are 2.29 for the a-N-CDs sample and 2.14 for the bass-CDs, describing quite smooth surfaces. The low roughness of the clusters’ surface indicates that they consist of small primary particles, with low porosity and a smooth surface.

Furthermore, SAXS measurements showed that the bass-CDs sample is characterized by the presence of smaller aggregates with a mass fractal behavior, evidenced by the characteristic power law behavior in the middle Q region (from approx. 0.0045 to 0.045 Å^−1^) (More details in the Appendix A). The best fit, with a specific fractal model, confirmed their mass fractal behavior, indicating the presence of aggregates with size of (127 ± 8) nm. The presence of these agglomerates was confirmed also by SLS measurement, which revealed the presence of clusters with an average diameter of (120 ± 30) nm (see Appendix A). The used model for SAXS measurements gives also the radius of the primary particles composing the aggregates, that is (4 ± 1) nm. For a-N-CDs, SLS turned out to be an unsuitable technique. In fact, they absorb the majority of the incident radiation and no intensity is able to reach the detector. AFM measurements highlight the spherical shape of both bass-CDs and a-N-CDs. The bass-CDs sample showed nanoparticles and clusters of different sizes. In Figure 2a, visible clusters of hundreds of nanometers (center and bottom of the image) and agglomerates with a diameter around 120 nm (top left of the image) can be seen. In Figure 2b,c, single smaller spherical nanoparticles with a diameter around 10 nm are shown. The nanoparticle diameters estimated by AFM are in accordance with the ones measured by SAXS and SLS. It appears that the a-N-CDs have a more homogeneous size distribution compared to the bass-CDs, with nanoparticles having a diameter of 13 nm (Figure 2d,e).

The optical properties of the CDs were investigated by Ultraviolet–Visible (UV–Vis) spectroscopy, Photoluminescence (PL) and time-resolved PL measurements.

UV–Vis absorption spectroscopy of aqueous solution (0.25 mg/mL) of bass-CDs exhibits a characteristic absorption peak around 250 nm, which is mainly derived from the π-π* transition of aromatic carbons. The other peak around 350 nm could be ascribable to the n-π* transition of double bonds containing groups such as C=O and C=N (see Appendix A) [40]. As seen in Figure 3, both bass-CDs and a-N-CDs have the same absorption peaks but the absorbances are far more intense in citric acid-derived nanoparticles.

The PL-PLE analysis of the bass-CDs revealed an excitation-dependent emission around 450 nm (Appendix A), with a decrease in the emission wavelength decreasing the excitation wavelength. This could probably highlight that the emission is from defect and not from well-defined electronic levels. For a-N-CDs, the high presence of one fluorophore gave a very stable and excitation-independent emission peak at 450 nm.

To fully explain the photocatalytic reactions, time-resolved Photoluminescence (PL) was applied to evaluate the PL lifetimes of the carbon nanoparticles (Figure 4). The average lifetime of the excited state (τ) of bass-CDs was 7.0 ns, while for a-N-CDs τ = 13.5 ns was observed. The τ of the dialyzed sample of bass-CDs was 6.5 ns, similar to the one of non-dialyzed bass-CDs. To fit the emission of bass-CDs, three constants were needed, while for a-N-CDs, only two were needed; the behaviors of the PL decays and the parameters are reported in Appendix A. The more linear character of a-N-CDs was consonant with a molecular-like emission, explained by the presence in its structure of molecular fluorophores [24].

Quantum Yields (QYs, based on quinine sulphate) were also measured for the three types of CDs (a-N-CDs, bass-CDs and dialyzed bass-CDs). In Figure 5, the spectra collected in the integration sphere are reported. The excitation wavelength is 372 nm. The blue, red, green and yellow curves refer to the scattering from a blank reference (Sr) and the sample (Ss), the emission from the reference (Er) and the sample (Es), respectively. The quantum yield is calculated using the following equation: QY(η) = (Es − Er)/(Sr − Ss).

The QYs for a-N-CDs, bass-CDs and dialyzed bass-CDs were, respectively, 17.3%, 6.0% and 3.1%. The large QY observed for the a-N-CDs was attributed to the presence of molecular fluorophores, while the purified bass-CDs have a ~48% lower QY, probably due to the loss of fluorophores.

Another parameter that was taken into account was the mass absorption coefficient ε. From our previous work, ε was dependent on the presence of nitrogen [24]; however, in this study ε = 10.93 L·g^−1^·cm^−1^ for a-N-CDs and ε = 0.37 L·g^−1^·cm^−1^ for the bass-CDs. These experimental data highlighted that ε was independent from the N-doping of the CDs: Although the C/N ratio is almost the same (3.23 for a-N-CDs and 3.21 bass-CD), the mass absorption coefficient is very different.

For easier comparison, in Table 3 the average lifetime (τ) of the excited state, the QY, and ε for the CDs, together with the photocatalytic activities (initial rate and conversion), are summarized.

The observed trend indicates that higher QY of a-N-CDs (17.3%) with respect to the bass-CDs (6.0%) does not strictly correlate with the kinetics (4.9 Ms^−1^ for a-N-CDs and 7.5 Ms^−1^ for a-N-CDs). This apparent inconsistency was explained because QY, ε and τ are helpful in understanding PET but not strictly correlated with the catalytic activity that depends on multiple variables.

### 3.2. Photocatalysis

Starting from the assumption that the photocatalytic activity of the CDs (i.e., their photoelectron transfer (PET) ability) depends on their morphology, structure and optical properties, we decided to probe it by studying the photoreduction of methyl viologen MV^2+^ to its radical cation MV^+^. In this reaction, an electron is transferred from the photoexcited state of the CDs to the MV^2+^ that acts as acceptor molecule and is an established reactivity-test to prove the PET efficiency of CDs [19,20,41] (Figure 6). To understand the photoredox activity and correlate it with the structure of the bass-CDs, we measured the photoreduction rates of MV^2+^ (−0.45 V vs NHE) in aqueous solution in the presence of ethylenediaminetetraacetate as a sacrificial electron donor under light-emitting diode irradiation at 365 nm (the time resolved UV-Vis spectra are shown in Appendix A) and compared with the one obtained with the already known a-N-CDs [24]. To compare the photocatalytic activity of two different types of CDs, we decided to perform two different experiments: One using a concentration of nanoparticles normalized for absorption at 365 nm and another using the same concentration (mg of CDs/mL) of photocatalyst.

In the experiment with the normalized absorption of CDs, bass-CDs resulted to be far less luminescent and a higher concentration of CDs was needed to reach the same absorption as a-N-CDs (0.2 mg/mL of bass-CDs vs 0.005 mg/mL a-N-CDs to have an absorbance of 0.13 a.u.); however, as shown in Figure 7 and Table 3, bass-CDs exhibited a higher initial photoreduction rate (7.5 × 10^−8^ vs 4.9 × 10^−8^ Ms^−1^) and a higher MV^2+^ conversion (34.0 vs 21.8 µM). On the other hand, using the same concentration of photocatalyst, a-N-CDs resulted to be far more luminescent (2.9 vs 0.13 a.u. at 365 nm) and only slightly more reactive toward PET to MV^2+^. In fact, as highlighted in Figure 7 and Table 3, after 30 min, a-N-CDs gave a substrate conversion of 62.5% (37.5 µM) compared to the 56.6% of bass-CDs. This underlines the high PET ability of bass-CDs, despite their low absorbance at 365 nm and without the need of any external nitrogen doping agent.

A series of control reactions with a-N-CDs (tested in the absence of light, without CDs and without EDTA) were already reported in our previous work [24] and demonstrated that, in the absence of irradiation, EDTA or CDs, no MV^2+^ reduction was observed, implying that all three were indispensable. Two additional control experiments, with the bass-CDs and in the absence of light (pink data in Figure 7) and with bass-CDS and without EDTA (green datapoints in Figure 7), further confirmed these requirements for the photochemical reduction of MV^2+^.

Cyclic Voltammetry (CV) experiments were carried out on a-N-CDs in our previous work [42] to measure their redox activity and correlate the energy levels of the nanomaterials with the relative band gap. a-N-CDs showed a negative potential (E onset red = −1.94 vs Ag|AgCl), suggesting a good reducing capability. The energy gap of the nanomaterials resulted to be of 3 eV, thanks to the presence of molecular-like fluorophores that ensure a high PET efficiency, implying a great energy of the excited electron generated upon irradiation.

## 4. Conclusions

In this work we demonstrate the possibility to valorize fishery waste, in particular bass scales, using them as a carbon source for the production of luminescent carbon nanoparticles with a high ability in photoelectron transfer. This new class of CDs was characterized in-depth regarding their morphology, composition and surface properties, showing a considerable nitrogen content (3.21 C/N from elemental analysis) without the need of any external doping agent. Since nitrogen doping in CDs is known to enhance the photocatalytic activity of the nanoparticles, bass-CDs were then compared with CDs obtained from citric acid and diethylenetriamine (a-N-CDs) toward the single electron reduction of MV^2+^.

Bass-CDs resulted to have a lower lifetime of the excited state (7.0 ns vs 13.0 ns of a-N-CDs), mass absorption coefficient (0.37 L·g^−1^ cm^−1^ vs 10.93) and QY (6.0% vs 17.3%), but higher initial photoreduction rate (7.5 × 10^−8^ M·s^−1^ vs 4.9 × 10^−8^ M·s^−1^ for absorption normalized experiment). This experimental evidence contributes to highlight the fact that τ, ε and QY are important parameters to understand the photoelectron transfer activity of CDs, but they are not strictly correlated with the photocatalytic ability of the nanoparticles. However, a lower value of QY seems to favor the photocatalytic activity of CDs, presumably because the absorbed photons are not immediately re-emitted but can instead transfer their energy to perform the photoreduction.

The fish scale-derived CDs have similar morphology to that of the a-N-CDs, but less intense absorbance, lower QY, τ and ε. Despite this, bass-CDs resulted to function as a valuable photocatalyst and to have a higher photoelectron transfer ability toward methyl viologen.

## Figures and Tables

**Figure 1 nanomaterials-11-00524-f001:**
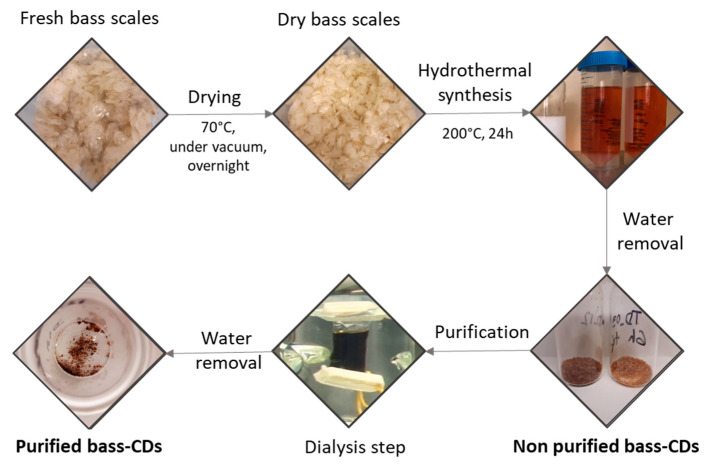
Synthetic pathway for the bass carbon dots (bass-CDs).

**Figure 2 nanomaterials-11-00524-f002:**
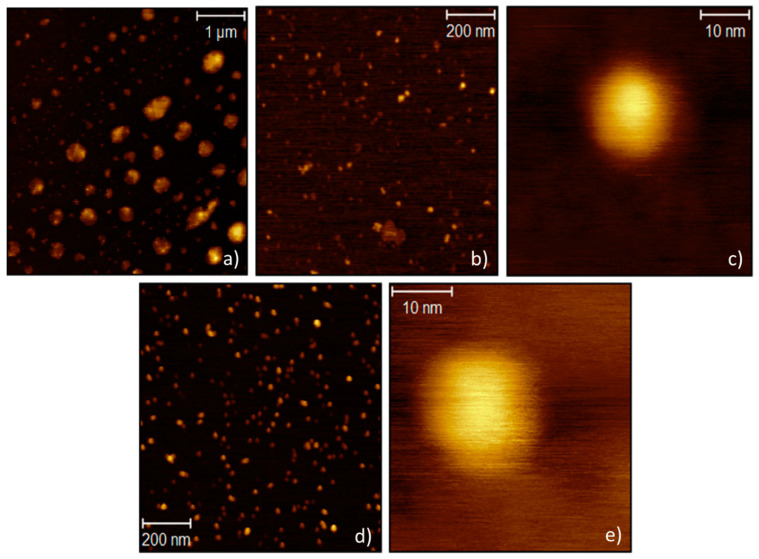
AFM images of bass-CDs (**a–c**) and a-N-CDs (**d,e**).

**Figure 3 nanomaterials-11-00524-f003:**
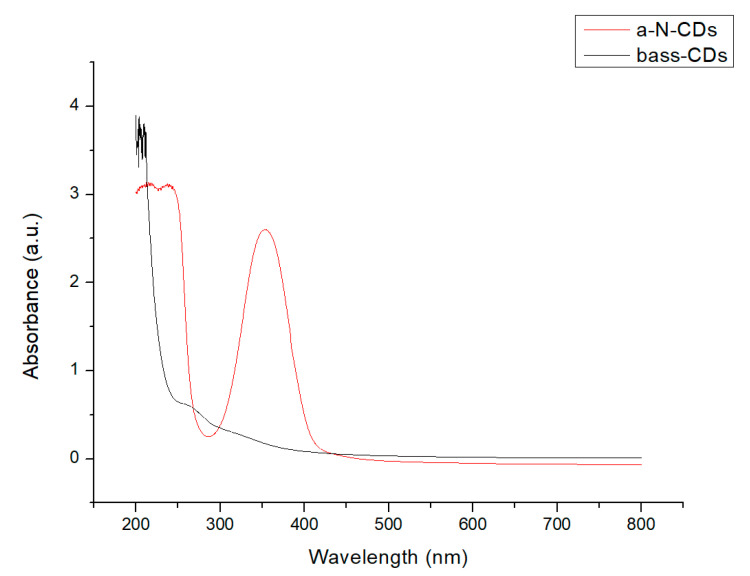
Comparison between UV–Vis spectra of bass-CDs (black line) and a-N-CDs (red-line).

**Figure 4 nanomaterials-11-00524-f004:**
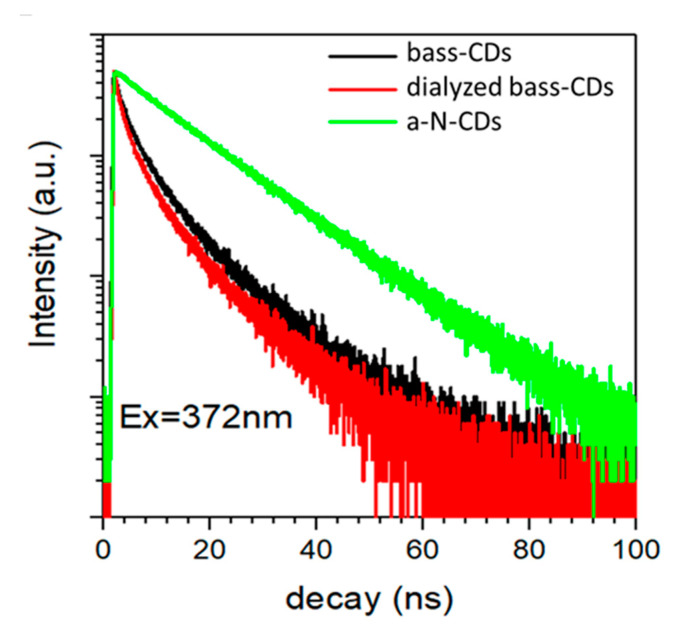
Time-resolved PL measurements of a-N-CDs (green), bass-CDs (black) and dialyzed bass-CDs (red).

**Figure 5 nanomaterials-11-00524-f005:**
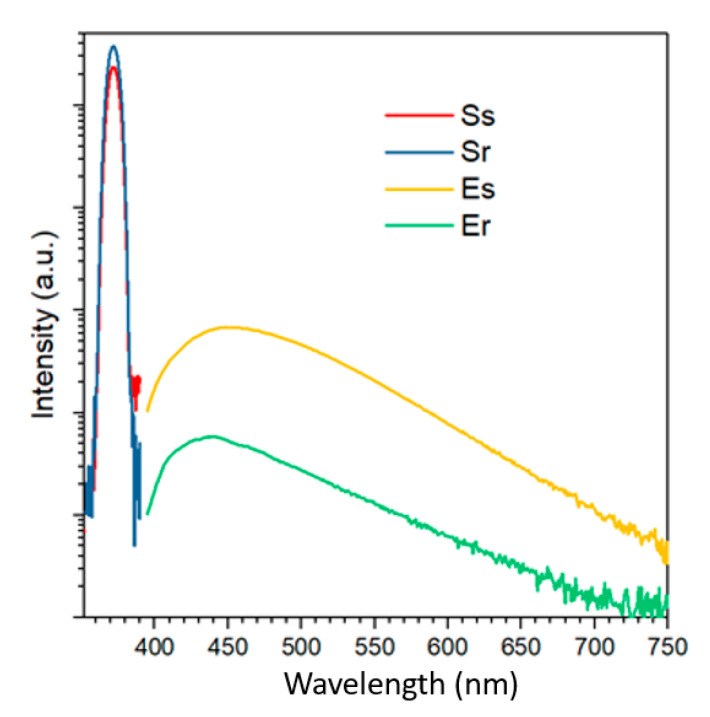
The PL spectra collected in the integration sphere.

**Figure 6 nanomaterials-11-00524-f006:**
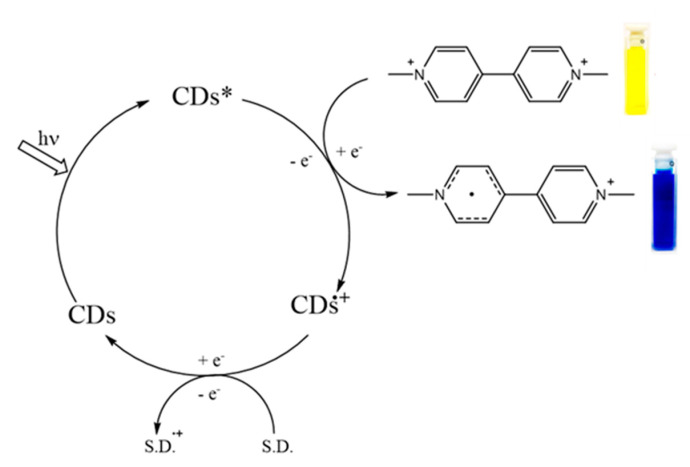
Catalytic scheme for the single electron photoreduction of methyl viologen with carbon dots (CDs).

**Figure 7 nanomaterials-11-00524-f007:**
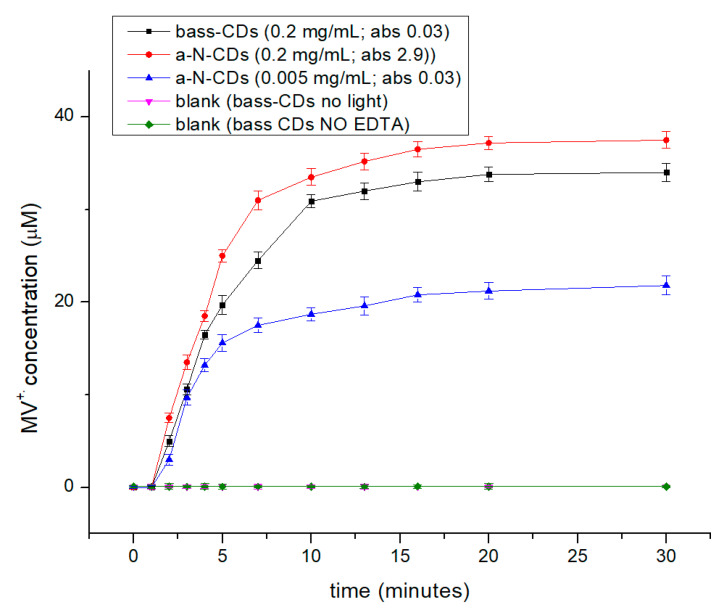
Reaction kinetics of the formation of methyl viologen radical (MV^+^) using bass-CDs (black datapoints) and a-N-CDs (red and blue datapoints). Blank tests in absence of light (pink datapoints) and of EDTA (green datapoints).

**Table 1 nanomaterials-11-00524-t001:** Elemental percentage composition of bass scales, bass-CDs and amorphous nitrogen-doped carbon dots (a-N-CDs).

	%N	%C	%H	%S	C/N
Bass scales	9.00	25.00	4.50	0.20	2.78
Bass-CDs	13.78	44.24	7.05	0.77	3.21
a-N-CDs	16.50	53.30	5.84	0.17	3.23

**Table 2 nanomaterials-11-00524-t002:** Comparison of morphology, composition and surface properties of bass-CDs and a-N-CDs.

	Elemental Analysis	FT-IR	XPS
Bass-CDs	C/N = 3.21	-OH -CH -NH (amines/amides) -carbonyl -Aromatic C=C	-C=C -C–O or C–N -C=O or C=N -NH_2_ -amides C/O/N = 66/26/8
a-N-CDs	C/N = 3.23	-OH -CH -NH (amines/amides) -ammonium -carbonyl -Aromatic C=C	-C=C -C–O or C–N -C=O or C=N -NH_2_ -C–N-C C/O/N = 70/20/10

**Table 3 nanomaterials-11-00524-t003:** Initial photoreduction rate (v_0_), Quantum Yield (QY), mass absorption coefficient (ε), average lifetime (τ), and MV^2+^ conversion of the synthesized CDs.

	CDs Concentration (mg/mL)	Absorbance at 365 nm	v_0_·10^−8^ (M·s^−1^)	QY (%)	ε (L·g^−1^ ·cm^−1^)	τ (ns)	MV^2+^ Conversion (%)
Bass-CDs	0.2	0.13	7.5	6.0	0.37	7.0	56.6
a-N-CDs	0.005	0.13	4.9	17.3	10.93	13.0	36.3
a-N-CDs	0.2	2.9	9.6	17.3	10.93	13.0	62.5

## Data Availability

Data is contained within the article or Appendix A.

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
