# Peer review of "Biobased Carbon Dots: From Fish Scales to Photocatalysis"

_nanomaterials, 2021, doi:10.3390/nano11020524_

Round 1

Reviewer 1 Report

This manuscript presented a new class of carbon dots made from fish scales and the photocatalytic performance is investigated. This is a feasible method to convert biowaste derives (fish scales) into useful materials. Materials characterizations and discussion are well presented. However, there are still a few points need to be improved, as listed below. 

  1. The introduction can be revised and reorganized in a better way to show the background and recent progress. 
  2. A few microscope images are recommended to show the morphologies of CDs. 
  3. In figure 1, the unit for wavelength should be nm. There are typo (wavelength) in the SI as well. 
  4. In Figure 3, the x-axis label should be wavelength. 
  5. What are the rest components in CDs besides C, N, H, S? Also, as the authors mentioned, fish scales contain a large amount of CaCO3. Did the authors remove the CaCO3 residual in CDs? 
  6. XRD of CDs are recommended to add.  
  7. The absorption intensity of bass-CDs at 365nm is around 0.4, based on Figure 1. The concentration was shown to be 0.25 mg/mL in line 258. However, in the catalytic result (table 3), the absorbance at 365 nm was only 0.03 when the bass-CDs concentration is 0.2 mg/mL. Can the authors confirm the concentration and absorbance? 
  8. More photocatalysis results are needed, such as blank controls, recyclability, photocurrent reponse, etc. Also, standard derivation is needed in figure 5. 
  9. The photocatalysis using normalized absorption of CDs is not a solid way to compare catalytic performance. 

Reviewer 2 Report

This work focuses on the synthesis, characterization and photoreduction ability of a new class of carbon dots made from fish scales. More specifically, self-nitrogen-doped carbonaceous nanostructured photocatalyst were synthesized from fish scales by a hydrothermal method in the absence of any other reagents. The morphology, structure and optical properties of these materials were investigated. Their photocatalytic activity was compared with the one of conventional nitrogen-doped carbon dots made from citric acid and diethylenetriamine in the photoreduction reaction of methyl viologen.

This is indeed an interesting work.

Nevertheless some minor revisions are needed in order to be published in MDPI Nanomaterials:

- A photograph of the as grown CDs is needed. Please present a SEM photograph.

- A flowchart or scheme showing the synthesis of the CDs would be helpful.

- I suggest the authors to recover their CDs and reuse them against the photoreduction of methyl viologen. the re-usability (for at least 3 times) of this catalysts is quite essential for real life applications.

- Are there similar works in the literature? Some new references are needed..

This is a novel work and can be published after covering the minor revisions mentioned above.

Author Response

PLease see the attcahment
